# Cryo-EM structure of transmembrane AAA+ protease FtsH in the ADP state

Wu Liu[1,2,5], Martien Schoonen[1,4,5], Tong Wang [3✉], Sean McSweeney[2] & Qun Liu [1,2✉]

AAA+ proteases regulate numerous physiological and cellular processes through tightly regulated proteolytic cleavage of protein substrates driven by ATP hydrolysis. FtsH is the only known family of membrane-anchored AAA+ proteases essential for membrane protein quality control. Although a spiral staircase rotation mechanism for substrate translocation across the FtsH pore has been proposed, the detailed conformational changes among various states have not been clear due to absence of FtsH structures in these states. We report here the cryo-EM structure for *Thermotoga maritima* FtsH (*Tm*FtsH) in a fully ADP-bound symmetric state. Comparisons of the ADP-state structure with its apo-state and a substrate-engaged yeast YME1 structure show conformational changes in the ATPase domains, rather than the protease domains. A reconstruction of the full-length *Tm*FtsH provides structural insights for the dynamic transmembrane and the periplasmic domains. Our structural analyses expand the understanding of conformational switches between different nucleotide states in ATP hydrolysis by FtsH.

[1] Biology Department, Brookhaven National Laboratory, Upton, NY, USA. [2] NSLS-II, Brookhaven National Laboratory, Upton, NY, USA. [3] Advanced Science Research Center at The Graduate Center, The City University of New York, New York, NY, USA. [4] Present address: Department of Bioengineering, Lehigh University, Bethlehem, PA, USA. [5] These authors contributed equally: Wu Liu, Martien Schoonen. ✉email: twang1@gc.cuny.edu; qunliu@bnl.gov

Protein quality control is a major cellular pathway to assure protein homeostasis and their regulated physiological and cellular activities. AAA+ proteases are a family of proteases that proteolytically cleave substrates from which the substrates are unfolded and delivered to the cleavage sites driven by ATP hydrolysis[1,2]. Among these AAA+ proteases, FtsH (Filamentation Temperature Sensitive Protein H) is the only known family anchored on membranes where it is able to hydrolyze both soluble and membrane protein substrates[3,4]. The quality control and regulation of membrane proteins by FtsH play a critical role in cell viability and stress resistance in response to changing environments.

FtsH is ubiquitously expressed in prokaryotic and eukaryotic organisms. In bacteria, FtsH proteolytically controls the protein quality of many soluble and membrane proteins[5]. Among these are LpxC in regulating membrane outleaf lipopolysaccharide (LPS) biosynthesis and homeostasis[6], YfgM in mediating stress response, YccA in regulating membrane protein folding[7], and transcription factor $\sigma^{32}$ in heat-shock response[8]. The proteolysis by FtsH is tightly regulated to assure timely and sensitive response of cells to changing environments. For example, the FtsH cleavage of LpxC is further regulated by the recruitment of two additional proteins YciM as a facilitator and YejM as an inhibitor[9–12]. Eukaryotic FtsHs mainly localize to organelles mitochondria and chloroplasts of prokaryotic origin. In mitochondrial inner membranes, FtsHs have their catalytic domains facing either the cell matrix or interspace, and thus can cleave substrates in either location[13,14]. In chloroplasts, the well-characterized function of FtsH is to repair the photosystem II (PSII) damage-caused photoinhibition through cleavage of the PS-II reaction center protein D1 subunit[15,16]. In addition to repairing PSII, in some plants FtsH has lost protease activity while maintaining ATPase activity to drive the import of proteins into chloroplast stroma[17,18].

FtsHs contain four domains: N-terminal periplasmic and transmembrane domains and C-terminal ATPase and protease domains. Crystal structures for the ATPase and protease domains suggest that FtsH forms a hexamer with a double-ring structure[19–21]. The crystal structure of the apo-state TmFtsH (FtsH from *Thermotoga maritima*) catalytic domains displays a six-fold symmetry for both the protease and the ATPase domains. Upon ADP binding, the protease domains retain six-fold symmetry, while the six ADP-bound ATPase domains showed a two-fold or three-fold symmetry in their crystal structures[20,22,23]. The break of symmetry in the ATPase domains raised a puzzle of how the six-fold apo-state structure changes its conformation to reach the ADP state. It's noted that the ADP-state crystal structure used only the soluble catalytic domains that formed a monomer in solution[20]. It is possible that under crystallization conditions, intermolecular packing and lattice contacts caused the formation of 2/3-fold symmetry.

There is no crystal structure available for FtsH in an ATP-state; we presume this is due to the significant conformational changes associated with ATP binding and subsequent hydrolysis. Recently, a cryo-EM structure for the catalytic domains of a yeast homolog YME1 provided the first ATP-state structure with a substrate peptide trapped in the ATPase pore[24].

Taken as a whole, these FtsH structures indicate the involvement of ATP-binding in substrate translocation through a "spiral staircase" mechanism[25]. However, these structures do not explain the transition between an ADP-state in resting conditions and an ATP-state during substrate loading and translocation. In addition, except one low-resolution structure[26], the FtsH structures available were determined mostly for their catalytic domains, missing the transmembrane and periplasmic domains, raising questions to understand the structure and function of these two domains.

To answer these questions around the relationship of FtsH structure and function, we determined a cryo-EM structure of TmFtsH in an ADP-bound state that reveals conformational changes from the ADP-state to apo- and substrate-loaded ATP-states. A reconstruction of the full-length TmFtsH suggests dynamic features for the transmembrane and periplasmic domains.

## Results

**Formation of TmFtsH-MSP1D1 nanodiscs.** We overexpressed the full-length, protease deficient TmFtsH (Fig. 1a) H423Y mutant and purified it in detergents n-Dodecyl-B-D-Maltoside and N,N-Dimethyldodecylamine N-oxide (LDAO) and used negative staining to check particle quality. Neither preparation gave good particles based on electron microscopy analysis using negative staining. Therefore, we expressed a membrane scaffold protein MSP1D1 and reconstituted the purified TmFtsH into MSP1D1 nanodiscs in soybean polar lipids. We used

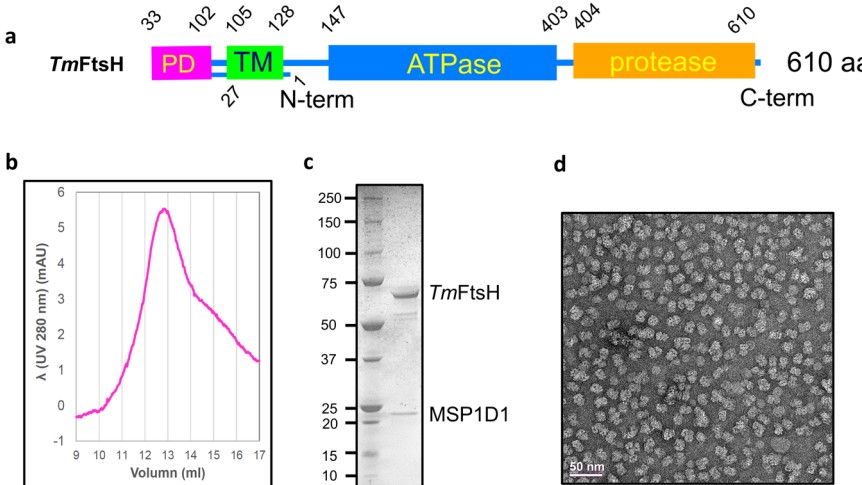

**Fig. 1 Production and purification, and characterization of TmFtsH. a** Schematic diagram of TmFtsH domains. PD periplasmic domain; TM transmembrane. **b** Size-exclusion chromatography and **c** SDS-PAGE analysis for the purified TmFtsH in MSP1D1 nanodiscs. **d** Negative-staining micrograph of TmFtsH reconstituted in MSP1D1 nanodiscs.

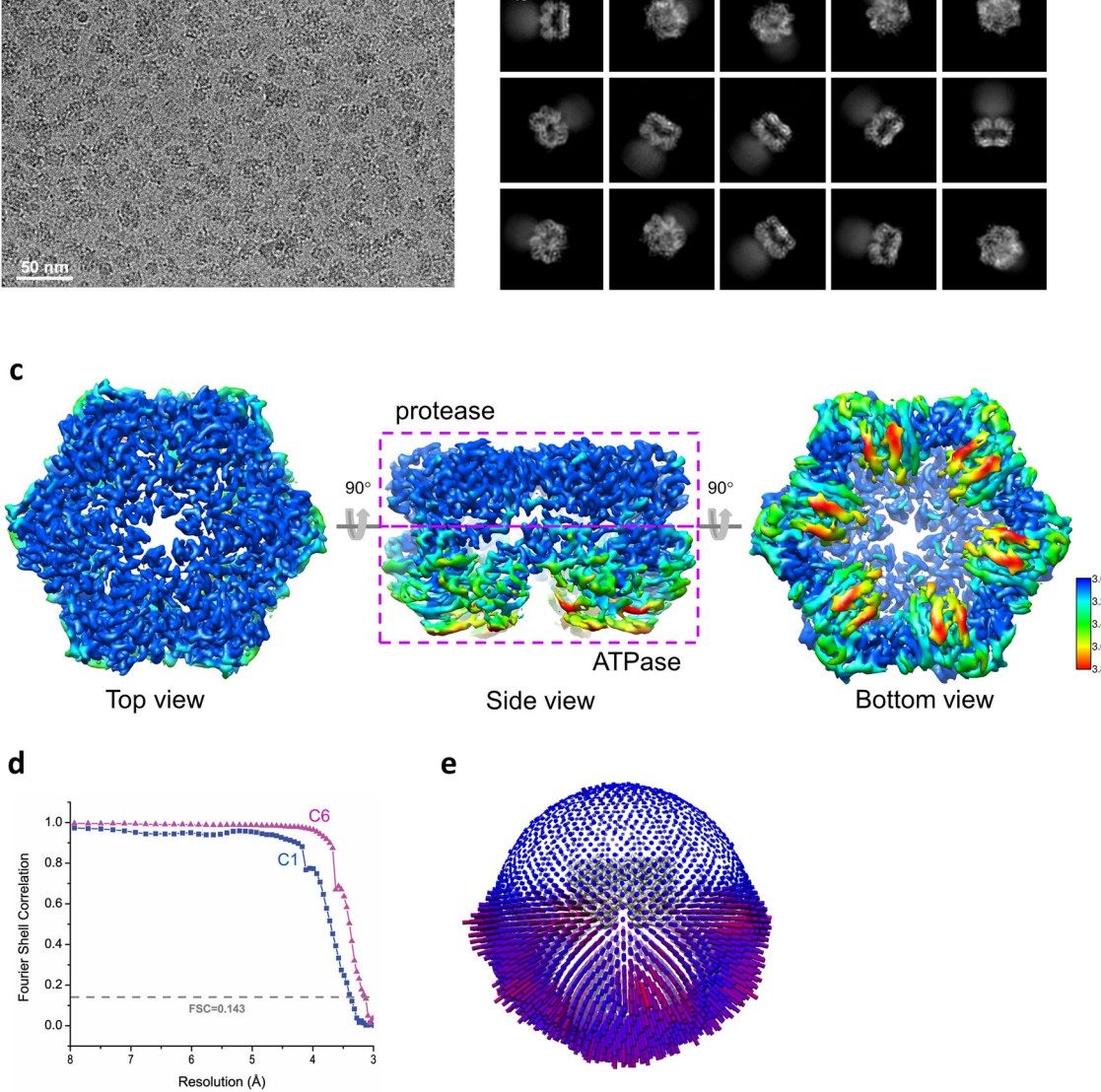

**Fig. 2 Determination of *Tm*FtsH structure in ADP state. a** A typical motion-corrected cryo-EM micrograph from a total of 11,169 micrographs. **b** 2D class averages. **c** Three views for the reconstructed map colored with local resolutions. **d** Fourier Shell Correlation (FSC) curve for the 3D reconstruction to determine the structure resolution. **e** Orientation distribution for particles used in the final 3D reconstruction.

size-exclusion column to polish the reconstituted particles and peak fractions containing *Tm*FtsH and MSP1D1 (Fig. 1b, c) were pooled together for structural analysis. Figure 1d shows a negative-staining electron micrography of *Tm*FtsH particles in MSP1D1 nanodiscs. These particles display multiple orientations and densities with no apparent aggregates.

**Structure determination**. For cryo-EM sample preparation, we tested various grids of C-flat, QuantiFoil, and QuantiAuFoil with different hole sizes and blotting conditions. All these preparations resulted in the denaturing of particles during the vitrification process. Only a few particles at the edge of each hole could be picked manually, resulting in an insufficient number of particles for subsequent data analysis. To enrich particles with less denaturation, we used QuantiFoil holey carbon grids coated with 2 nm ultra-thin carbon and added detergent CHAPS (3-cholamidopropyl dimethylammonio 1-propanesulfonate) to the sample prior to vitrification. Such treatment significantly improved the quality of particles, with suitable sizes and distribution, as indicated in Fig. 2a.

We determined the structure of *Tm*FtsH following the data analysis workflow as illustrated in Supplementary Fig. 1. 2D class averages show clear structural features for the ATPase and protease domains while with only faint densities for the transmembrane and periplasmic domains (Fig. 2b). Consequently, our final reconstruction at 3.15 Å resolution used 86,652 particles, consisting only the ATPase and protease domains. The reconstructed map used a six-fold symmetry and displays a higher resolution for the protease domains and lower resolution for the ATPase domains (Fig. 2c). As a comparison, a separate reconstruction with no symmetry (C1) yield a lower resolution of 3.4 Å (Fig. 2d). In the final reconstruction, most particles display side views, and fewer particles display views from the bottom (Fig. 2e).

The reconstructed map has a high quality and allows the building and refinement of atomic models. α-helices and β-strands can be clearly seen with characteristic sides chains to assist model building and refinement (Supplementary Fig. 2). Although we did not add ADP during the purification and sample preparation steps, we observed densities in the six ATPase

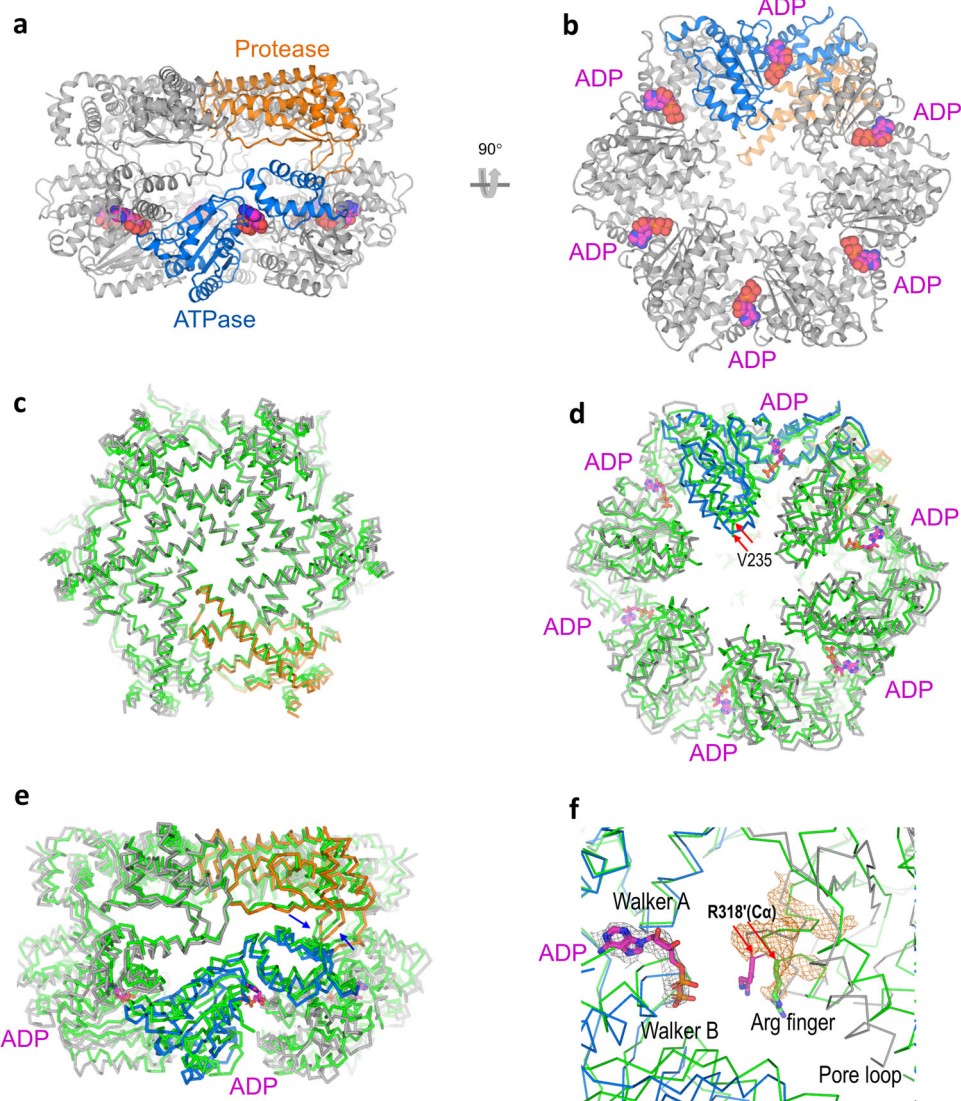

**Fig. 3 *Tm*FtsH structure in a symmetric ADP state. a**, **b** Two views of the overall structure of symmetric protease and ATPases domains. One subunit was colored orange for protease and marine for ATPase, and the rest subunits were colored in gray. ADP molecules are shown as spheres. **c**–**e** Three views of the superimposition of the ADP-state structure (colored in the same way as in **a** and **b**) with its apo-state crystal structure (PDB code 3KDS, colored in green). Both structures have a six-fold symmetry. The ADP-state structure was colored as in **a**, and the apo-state structure was colored in green. ADP molecules are shown as sticks. Red arrows indicate the Cα atom positions for measuring the distance between two V235 residues. Blue arrows indicate the loops connecting the protease and ATPase domains. **f** Superimposition of ADP-state structure and the apo structure for the binding sites as well as their next clockwise neighboring region. Arginine finger residue R318′ is shown as magenta (ADP-state) and green (apo-state). Red arrows indicated the Cα atom positions for measuring distance. Cryo-EM densities for ADP and the R318′ regions were colored as gray and orange isomeshes, respectively.

domains when we refined the map with or without symmetry (Supplementary Fig. 3). ADP bound endogenously to *Tm*FtsH and survived the co-purification and reconstitution processes. Our cryo-EM structure may thus represent a fully ADP-bound resting state structure in a six-fold symmetry under physiological conditions. As we used a zinc-binding-deficient mutant H423Y for cryo-EM analysis, we did not model zinc ions in the protease domains. Our refined model consists of 2484 residues and six ADP molecules (Fig. 3a, b).

**Apo-ADP state transition**. The apo-state *Tm*FtsH crystal structure for its catalytic domains has been previously determined with a six-fold symmetry[19]. We thus compared the apo- and ADP-state structures for seeking conformational changes induced by ADP binding. We attempted to align both structures based on overall, protease domains, and ATPase domains. We found that

the best alignment was based on the protease domains (residues 404–603) with an RMSD of 2.0 Å for 1128 aligned Cα atoms (Fig. 3c). With the alignment, the side view of the two hexamers indicated conformational changes in the ATPase domains starting from the linker loops to their protease domains (Fig. 3d, e and Supplementary Fig. 4a). Consequently, the bottom view indicated a shrink of the ATPase pore from the apo to ADP state by a movement of 4.5 Å as measured by the distance between the Cα atoms of residue V235 which is located on the tip of a pore loop (Fig. 3d).

Compared with the apo-state structure, ADP-binding triggered larger conformational changes for the Walker B region structure than the Walker A region structure (Fig. 3f). In addition, there are structural changes for the arginine finger in the next clockwise subunit where residue R318′ Cα atom ("R318′" denotes next clockwise subunit) moved 3.6 Å closer to the ADP phosphate

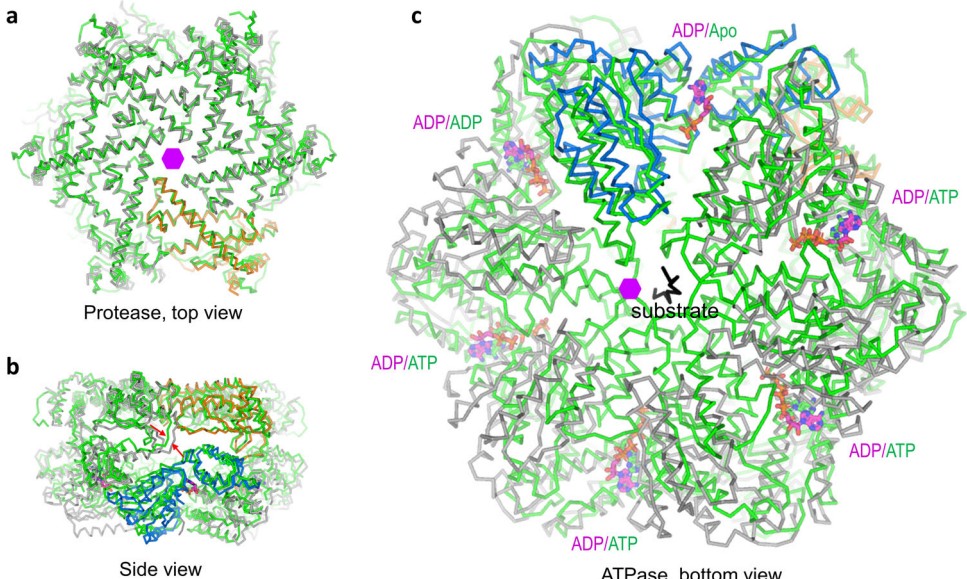

**Fig. 4 Structural comparison of ADP-state structure with a substrate loaded yeast homolog structure YME1. a** Top view to show the alignment for the protease domain. YME1 structure (PDB code 6AZ0) is colored in green; ADP-state *Tm*FtsH structure is colored in orange and gray. **b** Side view to show both protease and ATPase domains. Red arrows indicate the loops connecting the protease and ATPase domains. **c** Bottom view to show the alignment for the ATPase domains. ADP and ATP molecules are shown as sticks (magenta for *Tm*FtsH and green for YME1). Substrate is colored in black. Colors of the rest structures are the same as those of Fig. 3c–e. A magenta hexagon indicates the center of the symmetric ADP-state hexamer.

groups in the ADP-state structure (Fig. 3f). We observed that conformational changes in the arginine finger propagates to a disordered loop which is involved in substrate engagement and translocation (see below). However, we did not observe side chain densities for R318′. So, we used its mainchain Cα position and a sidechain rotamer to depict the residue position to illustrate structural changes. We propose that the R318′ sidechain amide groups may interact weakly with these negatively charged ADP phosphate groups, thus promoting a movement of the arginine finger closer to the ADP phosphate groups upon ADP binding. The apo-state structure is symmetric; ATP-binding and hydrolysis-induced conformational changes in the next clockwise subunit propagate throughout the hexamer. When all six ADP molecules bound in the hexamer, a new symmetric state, i.e. the ADP-state, is reached.

**ADP-ATP state transition**. To further understand the detailed conformational changes from ADP to ATP state prior to ATP-hydrolysis, we aligned *Tm*FtsH with a yeast homolog structure YME1 bound with four ATP molecules and a substrate peptide[24]. *Tm*FtsH and YME1 share a sequence identity of 37.2% (227 out of 610 residues). Similar to the alignment with the apo structure, *Tm*FtsH aligned well with the YME1 structure for the protease domains with an RMSD of 2.6 Å for 846 aligned Cα atoms (Fig. 4a). Conformational changes among the two structures are in the ATPase domains starting from the loops connecting their protease domains and are best viewed from the side of the two domains (Fig. 4b and Supplementary Fig. 4b). Compared with the protease domains, drastic conformational changes are seen in the ATPase domains (Fig. 4c). The YME1 hexamer contains four ATP, one ADP, and one empty site, thus providing a good reference to see conformational changes from the ADP- to ATP-state transition. Overall, ATP and ADP molecules in both structures overlap well. Upon ATP binding, its clockwise neighboring subunit changes its conformation, causing asymmetrical shrinkage of the pore relative to the center in the symmetric six-fold ADP-state structure (Fig. 4c).

Based on our structural analysis of the ADP-state with the apo *Tm*FtsH and substrate-loaded YME1 structures, it seems clear that ATP-binding and perhaps also substrate loading are required to break the six-fold symmetry. In absence of ATP, the conformational changes induced by ADP binding appear not big enough to create a tight pore for substrate engagement (Fig. 3d). We obtained the ADP-state structure without adding ADP or ATP during protein purification and cryo-EM sample preparation. This suggests that our ADP-state structure provides a basis to understand ATP-binding-dependent conformational changes.

**Comparison with the ADP-bound crystal structure**. The ADP-bound crystal structure of the *Tm*FtsH intracellular domains shows a mixed symmetry: six-fold for the protease domains and two-fold for the ATPase domains[23]. However, in our ADP-bound cryo-EM structure, we have both protease and ATPase domains in the six-fold symmetry. On the hexamer level, the two structures can only be aligned on the protease domains with an RMSD of 3.4 Å for 1010 aligned Cα atoms (Fig. 5a). In contrast, there are substantial conformational changes observed for the ATPase domains, and ADP molecules belonging to the same subunit did not align well (Fig. 5b). In the ADP-bound cryo-EM structure, the six ADP molecules are close to R318′ from their next clockwise neighboring subunits (Fig. 3f). As a comparison, in the ADP-bound crystal structure, four ADP molecules are close to their neighboring R318′; while the remaining two ADP molecules are further away from their neighboring R318′, suggesting very weak or no interactions (Fig. 5c).

In order to see detailed conformational changes between the two ADP-bound structures, we aligned one subunit of the cryo-EM structure with each of the three non-symmetric subunits in the crystal structure (Fig. 5d–f). With the alignments against the protease domain, conformational changes are in the ATPase domains starting on the loops (residues 403–410) connecting to the protease domain. These loops were ordered in the cryo-EM structure but disordered in the crystal structure for all subunits. In addition, in two of the three subunits, an 18-amino-acid loop

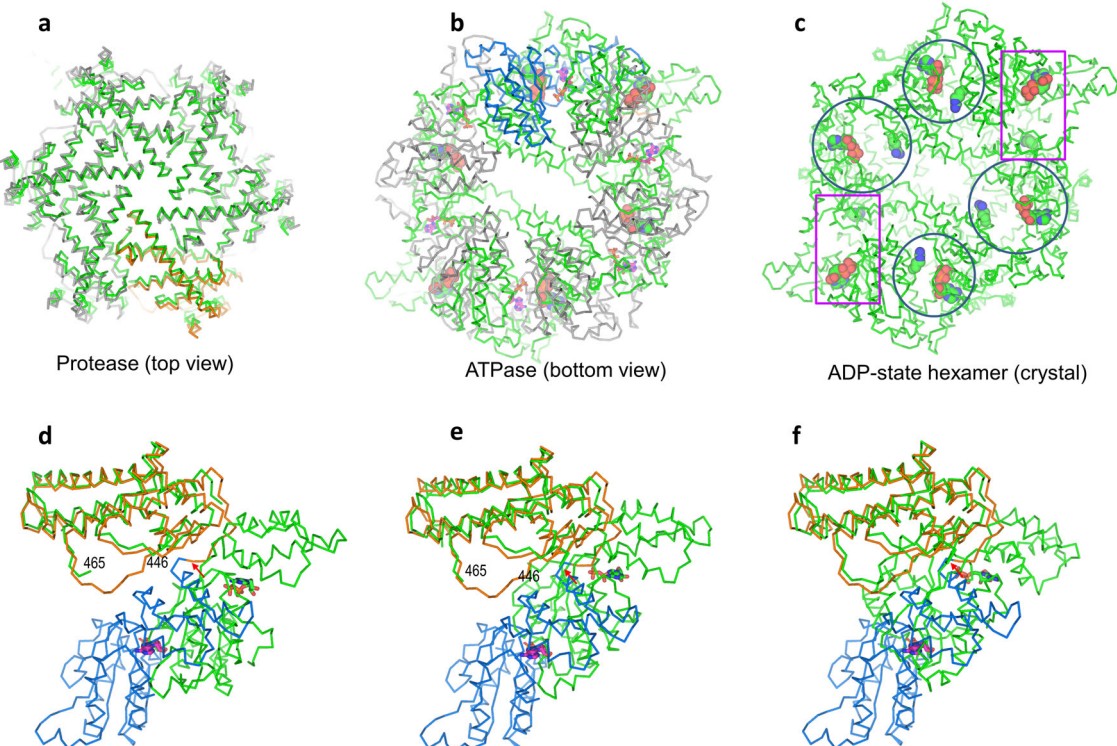

**Fig. 5 Structural comparison of ADP-state cryo-EM and crystal structures.** The crystal structure used for comparison is *Tm*FtsH intracellular domains (ATPase and protease) crystallized in a mixed C2/C6 symmetry (PDB code 2CEA). Two views to show the alignments for the protease (**a**) and ATPase (**b**) domains. ADP molecules are shown as sticks in cryo-EM structure and spheres in the crystal structure. Colors of the two structures are the same as in Fig. 3c–e. **c** Bottom view of the ADP-bound crystal structure for the ATPase domains to show the relative separation of ADP (orange spheres for carbon) and arginine finger residue R318 (green spheres for carbon). Circles and rectangular boxes indicate nearby and far away ADP-R318 pairs, respectively. **d–f** Side-view comparisons of one ADP-state subunit with three non-symmetric equivalents in the crystal structure. ADP molecules are shown as sticks: magenta for cryo-EM structure and green for crystal structure. Red arrows indicate the loops (disordered in the crystal structure) between the protease and ATPase domains.

in the protease domain (residues 447–464) is also disordered while the equivalent loops are ordered in the cryo-EM structure (Fig. 5d, e).

The ADP-bound crystal structure used only the catalytic domains (protease and ATPase) that formed a monomer in solution[20] whereas, our cryo-EM structure used the full-length protein. It is possible that under crystallization conditions, intermolecular packing and lattice contacts caused the formation of a distorted hexameric structure. We therefore argue that *Tm*FtsH in its fully bound ADP state should be interpreted as a six-fold symmetric structure for both the protease and ATPase domains.

**Full-length structure.** During our classification of the intracellular *Tm*FtsH hexamer, we found 2D classes that showed blurred transmembrane and periplasmic domains (Fig. 2b). We attributed this blurring to the denaturation of the two domains during the vitrification process. To obtain the full-length *Tm*FtsH reconstruction, we selected 2D classes generously and did extensive 2D classifications to find those rare 2D classes with best images for the transmembrane and periplasmic domains as shown in Fig. 6a. Through this 2D classification process, we were able to select 7468 particles, which mainly are side views. We excluded all top-view particles as it's difficult to see from projected views for the existence of transmembrane and periplasmic domains.

We then did ab initio 3D reconstruction, heterogenous, and homogenous refinements in CryoSPARC and obtained a reconstruction map at an overall resolution of 7.9 Å without symmetry (Fig. 6b, c). The local resolution map shows that the

intracellular domains have a higher resolution relative to the transmembrane and periplasmic domains (Fig. 6b). For the transmembrane domain, we could not detect α-helical features likely due to the disordering of the domain. Nevertheless, there are connecting densities between the transmembrane and periplasmic domains (Fig. 6b).

To produce a model to interpret the poorly defined transmembrane and periplasmic domains, we used AlphaFold[27] to assemble a hexameric model consisting of residues 1–134 and fitted the model into the density using Chimera (Fig. 6d, e). Relative to the intracellular protease and ATPase domains, the transmembrane and periplasmic domains tilted at the membrane surface by about 20°. Structurally, this tilted structure could be of biological relevance in allowing the proximity of substrates to the ATPase pore loops, thus promoting substrate recognition. Similar tilting has been suggested to be beneficial for the recognition and loading of membrane protein substrates[28].

## Discussion

There have been crystal structures for the soluble catalytic domains of the ADP-state structures for *Tm*FtsH and its bacterial homologs[20,22,29]. These structures were determined in a six-fold symmetry for the protease domains and two- or three-fold symmetry for the ATPase domains with fully loaded ADP molecules. Comparison of our ADP-bound cryo-EM structure with its crystal structure suggests that *Tm*FtsH ADP-state structure should be symmetric for both protease and ATPase domains (Fig. 5). Symmetric ADP-state cryo-EM structure provides a consistent picture to explain the conformational changes from

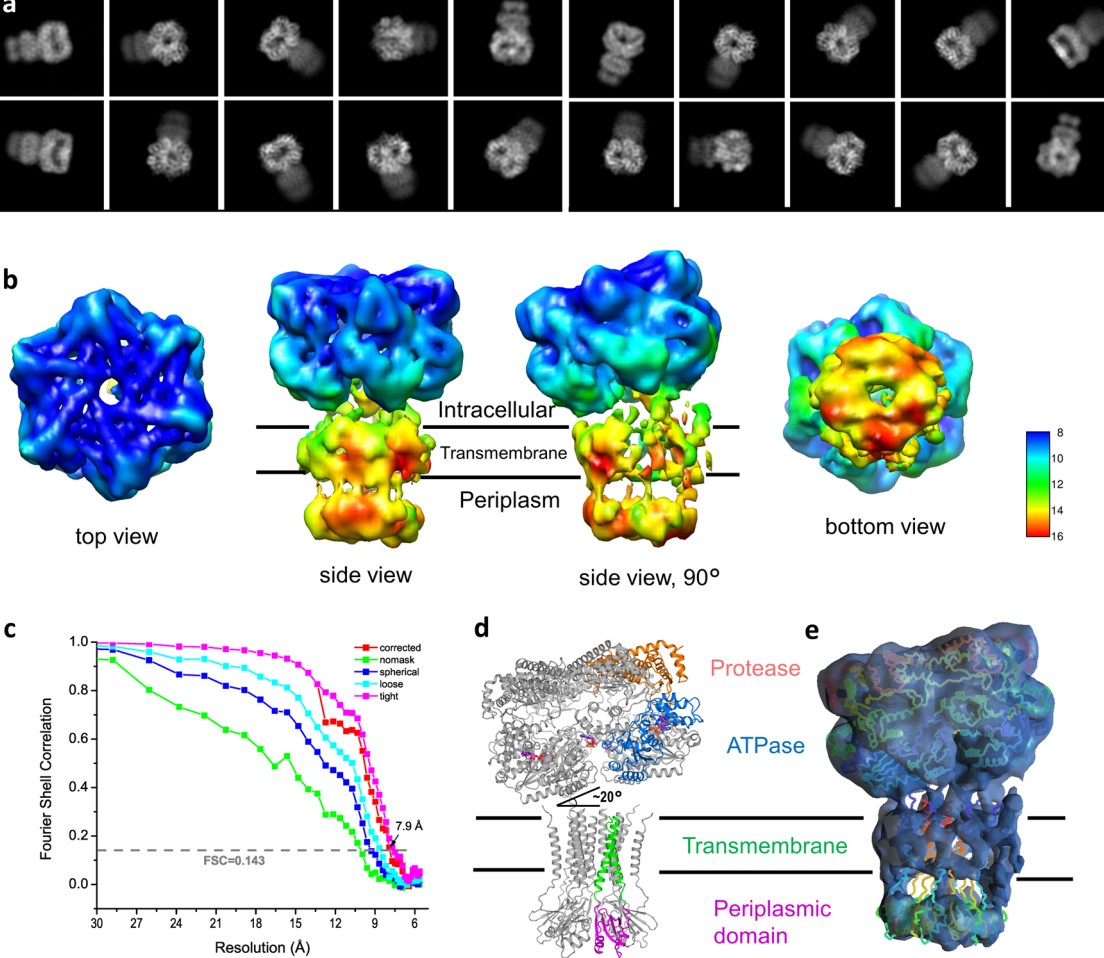

**Fig. 6 Full-length *Tm*FtsH reconstruction at a low resolution. a** Selected views of 2D class averages showing full-length *Tm*FtsH). **b** Four views of the reconstructed 3D map colored with local resolutions. **c** Gold-standard Fourier Shell Correlation. **d** A full-length *Tm*FtsH model with the transmembrane and periplasmic domains built by AlphaFold. **e** Fitting of the full-length *Tm*FtsH model into the density map.

apo to ADP-state through direct ADP-binding or ATP-hydrolysis induced movement of neighboring ATPase domains (Fig. 4d).

ATP-binding breaks the symmetry as shown in the YME1 structure[24]. We attempted to observe the breakage of *Tm*FtsH symmetry by ATP binding using cryo-EM. We added 5–10 mM ATP and 5 mM Mg$^{2+}$ to the protein and waited for 5–10 min before vitrification. We collected and processed cryo-EM data as we did for the ADP-state. However, the reconstructed structure remained in a symmetric ADP-state with six ADP molecules in the ATPase domains. It's possible that our protease deficient H423Y mutant[30] was active and hydrolyzed ATP to ADP. Using an ATP analog or a *Tm*FtsH mutant in the ATPase Walter B site might be possible solutions to obtain the *Tm*FtsH structure in its ATP state[31].

FtsHs have been proposed to form complexes with quite a few membrane protein substrates such as YccA[32], YfgM[33], YciM[11], and YqgP[34]. To get a high-resolution reconstruction for the transmembrane regions, one perhaps needs to have a well-formed complex between FtsH and its membrane protein substrate. Since ATP is needed for substrate engagement and translocation, the complex formation may also need ATP and a mutation in the ATP-binding site to slow down the ATP hydrolysis. This combined strategy will likely allow one to capture a high-resolution structure for the transmembrane and periplasmic domains in complex with a membrane protein substrate.

With the recently released AlphaFold algorithm for protein structure prediction, we were able to produce a hexameric model for the transmembrane and periplasmic domains to understand the low-resolution cryo-EM map for the full-length *Tm*FtsH. The model contains 12 transmembrane helices with loops connecting to periplasmic domains that form an oligomer in its crystal structure[35]. The predicted periplasmic domain structure is almost identical to its crystal structure with an RMSD of 1.3 Å for 200 aligned Cα atoms. However, the predicted transmembrane helices and their relative orientation to the periplasmic domains need to be further validated by a high-resolution structure.

The most well-studied function of FtsH is to cleave substrate proteins through which to control protein quality and homeostasis. As suggested by the full-length cryo-EM map, transmembrane and periplasmic domains are quite flexible compared to intracellular domains. Such a flexibility may be essential for their interactions with other proteins to acquire additional functions other than the protease function. One example is FtsH-inactive (FtsHi) proteins in plant chloroplasts where they form a large complex to import proteins into chloroplasts[17]. This function does not require the protease activity. It is thus possible that FtsH may form a translocation complex in which ATPase domains recognize and unfold substrate protein and deliver the unfolded protein for importing across the inner chloroplast membranes. In addition to FtsH, there are AAA+ protein

translocators that can indeed transport proteins across mitochondrial inner membranes[36,37].

With the ADP-state structure and its structural comparisons with the apo and substrate-loaded states, we could propose a process for the initial substrate loading and conformational changes. Under the resting conditions, FtsH is a symmetric hexamer, fully loaded with ADP in its six subunits. The intracellular protease and ATPase domains dynamically float over the membrane surface through the loops between the ATPase and transmembrane domains. A nearby substrate interacts to and is recognized by the ATPase pore loops. Such interactions are facilitated by tilting and the disordered loops between the ATPase and transmembrane proteins. For membrane protein substrates, hydrophobic interactions between their transmembrane domains may help to bring substrate close to the ATPase pore for facilitated substrate recognition and loading. When ATP is available, its binding triggers conformational changes of its clockwise next neighboring subunit which will break the six-fold symmetry and induce conformational changes to engage substrate. Subsequent ATP hydrolysis and translocation will follow the proposed spiral staircase mechanism[25] to pull the substrate through the ATPase pore and deliver it to the protease site for proteolytic cleavage.

## Methods

**Protein expression and purification.** The cDNA sequence encoding the full length of FtsH of *Thermotoga maritima* (*Tm*FtsH) was cloned into pET24-derived pMSCG-7 plasmid[38]. Using this plasmid as a template, FtsH H423Y mutation was generated by one-step site-directed mutagenesis[39]. After verification of the mutation by DNA sequencing, it was expressed in *Escherichia coli* BL21 (DE3) pLysS cells growing in Terrific Broth medium. After induction by 0.2 mM Isopropyl β-D-1-Thiogalactopyranoside (IPTG) at 16 °C for 18 h, the cells were harvested by centrifugation at 5000 rpm for 10 min using a Sorvall SLC-4000 rotor. Cell pellets were resuspended in lysis buffer (40 mM Tris, pH 8.0, 250 mM NaCl, 10% glycerol) supplemented by 1 mM phenylmethanesulfonylfluoride (PMSF, 100 mM stock in alcohol). Cells were lysed using an EmulsiFlex-C3 homogenizer (Avestin, Ottawa, Canada) at 10,000–15,000 psi for four passages. After another centrifugation at 16,000 rpm for 30 min using a Ti45 rotor, supernatant was collected, and the membrane was obtained by ultra-centrifugation at 36,000 rpm for 60 min using the Ti45 rotor.

Cell membranes were resuspended in solubilization buffer (40 mM Tris, pH 8.0, 250 mM NaCl, 10% glycerol, 0.5 mM TCEP, 1 mM AEBSF, 10 mM imidazole, 1.5% w/v LDAO, and Roche protease inhibitor cocktail) and stirred using a magnetic mixer. After 1 h, the solubilization mixture was ultra-centrifuged (Ti45 Rotor, 36,000 rpm) for 30 mins to remove the insoluble part. The supernatant was then passed through a pre-equilibrated HisTrap HP column (GE Healthcare, Chicago, USA) twice. The column was washed using a step gradient concentration of wash buffer (40 mM Tris, pH 8.0, 250 mM NaCl, 10% glycerol, 0.5 mM TCEP, 1 mM AEBSF, 0.06% w/v LDAO) supplemented with imidazole of 10, 50, 75, and 100 mM. The target protein was eluted using elution buffer (40 mM Tris, pH 8.0, 250 mM NaCl, 10% glycerol, 0.5 mM TCEP, 1 mM AEBSF, 0.06% w/v LDAO, and 300 mM imidazole). The eluted protein was polished by gel filtration on a Superose 6 10/300 increase column (GE Healthcare, Chicago, USA) with elution buffer of 20 mM HEPES, pH 7.6, 500 mM NaCl, 5% glycerol, 0.5 mM TCEP, and 0.06% w/v LDAO. Fractions containing the protein were pulled together and concentrated to ~1 mg/ml. Through the *Tm*FtsH solubilization and purification processes, ADP or ATP-Mg$^{2+}$ was not used in the buffer.

**MSP1D1 and lipids preparation.** Membrane scaffold protein MSP1D1 was expressed and purified from *E. coli* BL21 (DE3) pLysS cells[40]. Soybean polar lipids extracts (Avanti Polar Lipids, Inc., Birmingham, USA) were dissolved in chloroform and were transferred to glass tubes, dried under a nitrogen stream for 3 h. Residual chloroform was further removed by vacuum desiccation at 15 psi overnight, and the dried lipid films were resuspended in a buffer (100 mM sodium cholate, 20 mM HEPES, pH 7.6, 500 mM NaCl, 2 mM TCEP, 0.5 mM EDTA) and solicited for 1 hr in a bath sonicator until the liquid became clear. Finally, 50 mM soybean polar lipids were prepared in buffer containing 100 mM sodium cholate, 20 mM HEPES, pH 7.6, 500 mM NaCl, 2 mM TCEP, and 0.5 mM EDTA.

**TmFtsH-MSP nanodisc formation.** Reconstitution of *Tm*FtsH-MSP nanodiscs was based on the published method[40]. Briefly, purified protein in buffer A (20 mM HEPES, pH 7.6, 0.5 mM TCEP, 0.06% w/v LDAO) was mixed with buffer B (100 mM sodium cholate, 20 mM HEPES, pH7.6, 500 mM NaCl, 2 mM TCEP, 0.5 mM EDTA) to make a final concentration of sodium cholate at 25 mM. Then, soybean polar lipids and MSP1D1 were added to reach a final lipid concentration

of 0.32 mM and MSP1D1 concentration of 3.2 μM. The molar ratio of hexameric *Tm*FtsH:MSP1D1:lipid was 1:4:100. This mixture was incubated on ice for 1 h. Bio-beads SM2 (4 mg, Bio-Rad, Hercules, USA) were then added to the mixture, and the mixture was incubated at 4 °C for 1 h with constant rotation. A second batch of Bio-beads (8 mg) was added, and the mixture was incubated at 4 °C for 9 h with constant rotation. The reconstitution mixture was centrifuged and the supernatant was used for gel filtration through a Superose 6 increase column (GE Heathcare) in a buffer containing 20 mM HEPES, pH7.6, 150 mM NaCl, and 0.5 mM TCEP. The fractions containing the *Tm*FtsH-MSP nanodiscs were collected, concentrated to about 1 mg/ml, and flash frozen for storage at −80 °C. ADP or ATP-Mg$^{2+}$ was not used in the buffer for FtsH-MSP nanodisc formation.

**Negative stained sample preparation.** Glow-discharged continuous carbon grids (Cat# CF300-CU-UL, Electron Microscopy Sciences, Hatfiled, PA) were used for negative staining. The sample was diluted 10× in the elution buffer (20 mM HEPES, pH7.6, 150 mM NaCl, and 0.5 mM TCEP) and 3 μl was applied on the carbon side of the grid. After incubation for 1 min, extra protein was washed away by touching a piece of filter paper followed by two washes in 10 μl distilled water and one wash in 2% uranium acetate (UA). The sample was then stained using 2% UA for 1 min and excess liquid was wicked out using a filter paper. The stained sample was dried in air and evaluated on a JEOL-1400 Electron Microscope (JEOL USA inc, Peabody, USA).

**Cryo-EM sample preparation and data collection.** *Tm*FtsH-MSP1D1 nanodiscs were diluted by four times in a buffer containing 20 mM HEPES, pH 7.6, and 100 mM NaCl. The diluted sample was mixed briefly with CHAPS (3-((3-chola-midopropyl) dimethylammonio)-1-propanesulfonate) to a final CHAPS concentration of 0.05% (w/v). Three microliters of the mixed sample were immediately applied to a glow-discharged (15 mA current for 15 s) 300-mesh QUANTIFOIL R1.2/1.3 2 nm ultra-thin carbon grid (Cat# Q350CR1.3–2 nm, Electron Microscopy Sciences, Hatfiled, PA). After waiting for 60 s, vitrification was performed using a

**Table 1 Cryo-EM data collection, 3D reconstruction, and refinement statistics.**

| Data collection | |
| --- | --- |
| Microscope | Titan Krios G3i |
| Stage type | Autoloader |
| Voltage (kV) | 300 |
| Detector | Gatan K3 |
| Energy filter (eV) | 20 |
| Acquisition mode | Super-resolution |
| Physical pixel size (Å) | 0.684 |
| Defocus range (μm) | 0.7–2.5 |
| Electron exposure (e$^-$/Å$^2$) | 61 |
| *Reconstruction* | |
| Software | Relion v3.08, CryoSPARC v2.15 |
| Particles picked | 2,548,908 |
| Particles final | 86,652 |
| Extraction box size (pixels) | 400 |
| Rescaled box size (pixels) | 64 |
| Final pixel size | 1.368 |
| Map resolution (Å) | 3.15 |
| Map sharpending B-factor (Å$^2$) | 100 |
| *Model refinement* | |
| Software | PHENIX |
| Refinement algorithm | Real Space |
| Clipped box size (pixels) | None |
| Number of residues | 2484 |
| Number of ADP | 6 |
| *R.m.s deviations* | |
| Bond length (Å) | 0.003 |
| Bond angle (°) | 0.542 |
| Molprobity clashscore | 4.71 |
| Rotamer outliers (%) | 0.0 |
| Cβ deviations (%) | 0.0 |
| *Ramachandran plot* | |
| Favored (%) | 95.89 |
| Allowed (%) | 4.11 |
| Outliers (%) | 0 |
| PDB code | 7TDO |

ThermoFisher Mark IV vitrobot with a blotting condition of 3.5 s blot time, 0 blot force, and 100% humidity at 6 °C.

Cryo-EM data were collected with the use of a ThermoFisher Titan Krios (G3i) equipped with a Gatan K3 camera and a BioQuantum energy filter. With a physical pixel size of 0.684 Å, a total dose of 61 e$^-$/Å$^2$ were fractioned to 51 frames under the super-resolution mode using the ThermoFisher data acquisition program EPU. A total of 11,169 movies were collected with an energy filter width of 20 eV throughout the data acquisition. Data collection statistics are listed in Table 1.

**Cryo-EM data processing**. Beam-induced motion correction was performed using MotionCorr2[41] with a bin-factor of 2. Corrected and averaged micrographs were further corrected by CTF estimation using Gctf[42]. Micrographs with an estimated resolution better than 4.5 Å were selected for further processing. Particle picking and extraction were performed using Localpicker[43] and Relion3[44]. A total of 2,548,908 particles were picked, extracted at 400 pixels, and binned to 64 pixels with a pixel size of 4.49 Å.

We used CryoSPARC[45] and Relion3 for 2D and 3D class averages and 3D refinements. Specifically, we used 2D class averaging for initial particle cleanup which resulted in 842,697 particles. Using these particles, we produced an initial 3D map in CryoSPARC and used it to perform 3D homogenous refinements (Supplementary Fig. 1). Then these particles were selected, re-centered, re-extracted at 400 pixels, and binned to 200 pixels with a pixel size of 1.368 Å.

Extracted particles were auto refined to convergence in Relion3 followed by non-alignment 3D classification into five classes (Supplementary Fig. 1). Particles from the 3D class with the best structural feature as visualized in Chimera[46] were selected. Particles from the best class (15.4%) were selected for CTF refinement and Bayesian polishing in Relion3, and 2D classification and non-uniform refinement in CryoSPARC to reach a refined reconstruction based on gold-standard Fourier Shell Correlation of 0.143 (Fig. 2d). The final reconstruction used 86,652 particles. Local resolutions were estimated using BlocRes[47]. Reconstruction statistics are listed in Table 1.

**Model building and refinement**. To assist our model building and refinement, we sharpened the masked and filtered map using PHENIX[48] with a B factor of −100 Å$^2$. We used the PDB code 3KDS as a starting model and built the model for *Tm*FtsH and ADP in COOT[49] and refined the model iteratively in PHENIX. The refined model was validated using Molprobity[50] and the refinement statistics are listed in Table 1. The model for the hexameric transmembrane and periplasmic domains were built using ColabFold[51], a modified version of AlphaFold[27], and was fitted into the cryo-EM map as a rigid body in Chimera.

**Reporting summary**. Further information on research design is available in the Nature Research Reporting Summary linked to this article.

## Data availability

The three-dimentional cryo-EM density map has been deposited in the Electron Microscopy Data Bank under the accession number EMD-25837. Atomic coordinates have been deposited in the Protein Data Bank under the accession number 7TDO. Source data for Fig. 1c are available in Supplementary Data 1.

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

## Acknowledgements
We thank Huilin Li for helpful discussions and suggestions, Minge Du and Zuanning Yuan for screening cryo-EM samples, LBMS staff for the help with the cryo-EM operation and data acquisition. Q.L. was supported by the U.S. Department of Energy (DOE), Office of Biological and Environmental Research as part of the Quantitative Plant Science Initiative at Brookhaven National Laboratory. T.W. was supported in part by a PSC-CUNY Award 64507-00 52, jointly funded by the Professional Staff Congress and the City University of New York. The work used Laboratory for Biomolecular Structure (LBMS) which is supported by the U.S. Department of Energy, Office of Science, Office of Biological and Environmental Research.

## Author contributions
T.W. and Q.L. designed the study and experiments. W.L., T.W., M.S., and Q.L. performed the experiments. T.W., S.M., and Q. L. analyzed the data. Q.L. wrote the manuscript with help from other coauthors.

## Competing interests
The authors declare no competing interests.
