## [Peer Review File · Communications Biology]

Reviewers' comments:

Reviewer #1 (Remarks to the Author):

Evaluation of the manuscript "Cryo-EM Structure of Transmembrane AAA+ Protease FtsH in the ADP State" by Wu Liu et al.

The manuscript describes the cryo-EM structure of the AAA-protease FtsH in the ADP state. FtsH is an important membrane embedded AAA-ATPase with protease domain that manages degradation of membrane associated proteins in prokaryotes, where it is for example involved in lipid regulation. Ortholog proteins are present in mitochondria and chloroplasts of eukaryotic cells where they are especially important as safeguards for the energy generating systems. Due to the membrane embedded nature of the protein, structural information about the protein is sparse and largely based on segments of the full-length protein. The manuscript of Wu et al, therefore represents an important contribution to our understanding of the functioning of this important protein. The data allowed comparison of the ADP bound form to the apo- and ATP loaded forms and provide a glimpse on the N-domain and trans-membrane domain of the full-length protein.

Overall, the manuscript is well written, and most of the figures are of high quality. However, there are some minor issues that should be addressed to improve the manuscript:

P3, line 13: The writing suggests that the authors purified the wild-type protein, which is however not the case. The authors should clearly state already at the beginning of the results section that they purified a protease deficient (H423Y) mutant.

P4, line 31: ("'" denotes next clockwise subunit) should better read (" R318' " denotes next clockwise subunit)

P4, lines 6 and 7 need grammar improvement ("have views from the top"; "much few particles").

P5, line 1: It is no clear to this reviewer how the authors could depict the arginine 318 in Fig 3f when they did not see the side chain. Please clarify.

P5, line 10: some number for the similarity between scYme1 and TmFtsH would be helpful.

P5, line 19: needs grammar improvement

P5, line 25: Fig 3f does not fully show the pore loops. This has to be changed to be able to follow the authors arguments.

P6, line 1: needs grammar improvement

P6, line 20 and 21: Not clear what the authors mean with "disordered N-terminus of the protein itself". What is the evidence for this statement? Could these densities be parts of the ATPase domain not used for model building (i.e. prior to residue 158) or after the region used for AlphaFold modeling (i.e. after residue 134)?

Figures: The Figures are of good quality, but I suggest that the designation of the nucleotide (ADP) is moved outside the structure and written in the same color as the nucleotide is depicted in the model (similar as done in Fig. 4C).

Fig. 1a: PD and TM need also aa residue numbers.

Fig 3: Legend, line 6: (apo-sate) should read (apo-state)

Supplementary Fig. 4 legend: Here it is not clear which models were used for comparison. The pdb numbers and references of the used models have to be stated (also in other figures where relevant).

Reviewer #2 (Remarks to the Author):

Summary:

The authors have solved the structure of *Thermotoga maritima* (Tm) FtsH in an ADP-bound state using cryo-EM. The cryo-EM imaging was conducted using full-length TmFtsH, and the authors solved the ADP-bound structure comprising hexameric ATPase and protease domains. Through a

careful selection of 2D classes containing not only the ATPase and protease domains but also the transmembrane and periplasmic domains, the authors were also able to solve the structure of the ADP-bound full-length TmFtsH with low resolution. By comparing their ADP-bound structure with apo- or ATP-bound structures, conformational changes in the ATPase domains, rather than in the protease domains, were observed. When considering the structure of the full-length TmFtsH, the authors found tilted topologies of transmembrane and periplasmic domains with extra densities in the large gap formed by the tilting, suggesting the mechanism of substrate recognition and initial loading. However, the biological relevance of this tilted structure has not been well supported, and the extra densities in the tilted region have not been fully described, even in the figures. Additionally, while the authors mentioned that the conformational changes in the ATPase and protease domains begin on the connecting loops, there is no detailed description provided. Although the manuscript is well written and easy to follow, the authors need to address the specific concerns listed below.

1. Throughout the manuscript, the description in the text and figures is not completely matched (e.g., on page 4, line 4, Fig 3c needs to be replaced with Fig 2c). In addition, there are several typos in the figure legends (e.g., in figure 2, even though there is no panel "g," it is mentioned in the legend.).
2. Page 4, line 15: The authors have used a construct with H423Y mutation (a zinc-binding-deficient mutant) for cryo-EM analysis and have mentioned it as an active form in the discussion (page 7, line 5). Please show the additional data or provide pertinent explanation to support this.
3. On page 5, line 1, the authors have mentioned that side chain densities were not observed for Arg318', however, the side chain of Arg318' has been marked in figure 3f. In addition, the authors have proposed a potential interaction between Arg318' and phosphate groups in ADP, but no pertinent evidence, such as a distribution of charges in the neighboring region or additional densities around the phosphate groups, has been provided.
4. Page 5, line 28-29: How does this structure provide the basis for substrate loading and translocation processes?
5. In "Apo-ADP state transition" and "ADP-ATP state transition" sections, the authors have mentioned that the structural changes begin on the loops connecting the protease and ATPase domains. However, there is no description of how the structural changes in the loops are propagated to the ATPase domain.
6. In "Full-length FtsH structure" section, the authors have introduced the full-length structure of TmFtsH whose ATPase and protease domains are tilted relative to the lipid bilayer. Is this tilted structure biologically relevant?
7. In the same section, the authors have suggested that the extra densities in the large gap formed by the tilting could originate due to the substrate or the disordered N-terminus of the protein itself. However, considering the low resolution of the full-length structure, especially in the transmembrane domain, it is not clear whether it is reasonable to assume extra densities. It is recommended to provide an additional figure with a description regarding the extra densities.
8. Page 7, line 8: What is the supporting data for this hypothesis?
9. The authors have used the AlphaFold model structure to fit low-resolution densities in the full-length structure. However, it is unclear how the authors created a hexameric model structure using AlphaFold. Since the AlphaFold-Multimer was released recently, it would be helpful to mention in the method section, which version was used for the modeling and how it was carried out (AlphaFold, AlphaFold-Multimer, AlphaFold with modifications, or AlphaFold with Colab, etc.).

Minor Concerns:

1. Please systematize the usage of the abbreviated letter code for amino acids.

2. Page 4, 2nd paragraph: It would be better to match the flow of the text and order of figures for easy understanding.
3. Figure 1a: Please provide the explanation for PD and TM in the legend.
4. Figure 4: The meaning of cyan color should be noted.
5. Figure 5e: Please add the explanation for Purple colored arrow in the legend.
6. Supplementary Figure 1: Please add symmetry information for the refinement of the structure.

Reviewer #3 (Remarks to the Author):

The authors provide a short report of the cryoEM structure of the ATP-dependent protease *Thermotoga maritima* FtsH to 3.15 angstrom resolution. FtsH is a membrane-anchored protease that can degrade both membrane and soluble proteins, and the authors use in vitro reconstitution into lipid nanodiscs to isolate and characterize the full-length protein in the fully-ADP bound state. This resolution is a significant improvement on a recently published study (Carvalho et al, 2021) visualizing detergent solubilized full-length FtsH. Previous cryoEM structures of mitochondrial homologues of FtsH reported highly asymmetric interactions with both nucleotide and trapped polypeptide substrates. Here, the authors reveal a largely symmetric structure with all six subunits bound to ADP.

The visualization of the fully-ADP state is of moderate interest to researchers in the AAA+ enzyme field and the high resolution of the reconstruction will add another group of states that are available to be occupied by these enzymes. However, it is unclear under what circumstances the enzyme will occupy this fully-ADP state. The authors suggest this may represent a 'resting state' of the enzyme when ATP is not available, but it would be helpful if they could provide some evidence from the literature regarding the relevance of this state under physiological conditions.

The previous cryoEM structures of mitochondrial FtsH-like enzymes lack the transmembrane domains and small domains that sit on the other side of the membrane. In that regard, information on the structure of these domains in the full-length enzyme is useful, albeit these domains are flexible and poorly visualized in this structure. A higher resolution reconstruction of these regions would have made this structure far more interesting.

The microscopy is well-performed, and the validation of the structure is sufficient.

Specific points

1. The authors state that they applied 6-fold symmetry to the reconstruction and that an alternative reconstruction lacking any applied symmetry had slightly lower resolution. How do the positions of the ATPase domains in these two reconstructions compare? Is there evidence of asymmetry in the unconstrained map that is being lost when 6-fold symmetry is applied?
2. The manuscript contains a number of spelling and grammatical errors. It would be improved by a careful revision to remove these errors.

e.g. --Figure 1e is mislabeled in the legend.
--Page 3 Line 31 "followed" should be "following"
--Page 4 line 7 "much few" should be "many fewer"

Reviewer #4 (Remarks to the Author):

This paper reports a cryo-EM study of AAA+ protease TmFtsH in ADP-bound states, including a low-resolution reconstruction of the full-length complex. While the work appears to be technically sound, the biological insights obtained are limited and incremental. Particularly, there is lack of biochemical validation of the purified and imaged complexes, thus leaving it to be less well-defined in terms of the functional state of the solved structure. With additional verification and control, the

authors indicated that these structures represent the resting state, which is likely but still hypothetical, and weakly discussed the possible meaning of the structural model and how it adds to our knowledge to the system. In this reviewer's opinion, the paper may be enhanced at least by running a few basic assays to test the protease function. If possible, it will greatly help with a few control cryo-EM reconstructions with ATP or substrate added, which would make the structural interpretation and insights more reliable. At the current situation, one can only make bold assumption and vague discussion as to what has happened to the complex and what we can learn from the symmetric structure, when nearly all substrate-bound AAA+ ATPase complex showed asymmetric conformations. Additionally, the authors might want to consider the following issues when revising the paper.

- (1) The authors need to clarify the biochemical condition with respect to the nucleotide type and concentration used in the structural study both in the first section of results and in the Methods. Were the full-length TmFtsH purified with the presence or the absence of ADP or ATP in the buffer? Any magnesium ions provided in the buffer? If no, explain the rationale for why no nucleotides were supplied for the complex?
- (2) Where do the ADP ligands bound come from? Were they bound endogenously and copurified or added in the late step of sample preparation?
- (3) What were the nucleotide states of the full-length TmFtsH reconstruction shown in Fig. 5?
- (4) Have the authors conducted degradation assay and/or ATPase activity assay to verify the biochemical activity and function of FtsH or FtsH-MSP nanodisc?
- (5) The discussion of comparison of the ADP-bound cryo-EM structure with those of ADP-bound crystal structures is not clear. There appears to be notable differences in Supplementary Fig. S4. However, no discussion on such structural differences is provided and any explanation on the compatibility of cryo-EM and crystal structures, as well as why the two methods yield different conformations of the same nucleotide states. This concerns whether the current study rectifies the previous crystal structure study and whether the presented structure implicates a different mechanism.
- (6) Can the authors comment on the possible reason why all six subunits are in symmetric ADP-bound state? It also seems to be a simple test if the authors can add ATP to the purified complex to see if their assertion that ATP and substrate binding are needed to break the symmetric is correct.
- (7) While the authors claim in the abstract that a possible mechanism of substrate recognition and loading, there are no well-phrased passage describing such a mechanism, except for a few sentences speculating highly hypothetical, vaguely conveyed possibilities. These may not be sufficient to support their claim in the abstract.

Reviewers' comments:

Reviewer #1 (Remarks to the Author):

Evaluation of the manuscript "Cryo-EM Structure of Transmembrane AAA+ Protease FtsH in the ADP State" by Wu Liu et al.

The manuscript describes the cryo-EM structure of the AAA-protease FtsH in the ADP state. FtsH is an important membrane embedded AAA-ATPase with protease domain that manages degradation of membrane associated proteins in prokaryotes, where it is for example involved in lipid regulation. Ortholog proteins are present in mitochondria and chloroplasts of eukaryotic cells where they are especially important as safeguards for the energy generating systems. Due to the membrane embedded nature of the protein, structural information about the protein is sparse and largely based on segments of the full-length protein. The manuscript of Wu et al, therefore represents an important contribution to our understanding of the functioning of this important protein. The data allowed comparison of the ADP bound form to the apo- and ATP loaded forms and provide a glimpse on the N-domain and trans-membrane domain of the full-length protein.

Overall, the manuscript is well written, and most of the figures are of high quality. However, there are some minor issues that should be addressed to improve the manuscript:

Response: We thank this reviewer for positive comments on the significance and contribution of our work.

P3, line 13: The writing suggests that the authors purified the wild-type protein, which is however not the case. The authors should clearly state already at the beginning of the results section that they purified a protease deficient (H423Y) mutant.

Response: We have made this clear in the Results section on the use of the protease deficient H423Y mutant. (p. 3, line 20)

P4, line 31: ("'" denotes next clockwise subunit) should better read (" R318' " denotes next clockwise subunit)

Response: Corrected. (p. 5, line 9)

P4, lines 6 and 7 need grammar improvement ("have views from the top"; "much few particles").

Response: We revised the sentence to "In the final reconstruction, most particles display side views, and fewer particles display views from the bottom." (p. 4, lines 16-17)

P5, line 1: It is no clear to this reviewer how the authors could depict the arginine 318 in Fig 3f when they did not see the side chain. Please clarify.

Response: Due to no density for the R318 side chains, we used its mainchain C α position and sidechain rotamer to depict the residue. We have added this statement to the revised text. (p. 5, lines 12-14).

P5, line 10: some number for the similarity between scYme1 and TmFtsH would be helpful.

Response: Thanks for this great suggestion. *TmFtsH* and *YME1* share a sequence identity of 37.2% (227 out of 610 residues). We have included these numbers in the revision. (p. 5, lines 23-24).

P5, line 19: needs grammar improvement

Response: We revised the sentence to “Upon ATP binding, its clockwise neighboring subunit changes its conformation, causing asymmetrical shrinkage of the pore relative to that in the symmetric 6-fold ADP-state structure.” (p. 5, line 32 – p. 6, lines 1-2)

P5, line 25: Fig 3f does not fully show the pore loops. This has to be changed to be able to follow the authors arguments.

Response: Fig. 3d shows the pore loops. We there changed citation of Fig. 3f to Fig. 3d in the revised version. (p. 6, lines 5-7)

P6, line 1: needs grammar improvement

Response: We revised the sentence for better understanding. (p. 7, lines 6-8)

P6, line 20 and 21: Not clear what the authors mean with “disordered N-terminus of the protein itself”. What is the evidence for this statement? Could these densities be parts of the ATPase domain not used for model building (i.e. prior to residue 158) or after the region used for AlphaFold modeling (i.e. after residue 134)?

Response: Using “disordered N-terminus of the protein itself” has the same meaning as what the reviewer pointed out, i.e. prior to residue 158 or part of the ATPase domain. Due to uncertainty to the content of the extra density, we choose to delete this statement in the revised text. (p. 7, line 27).

Figures: The Figures are of good quality, but I suggest that the designation of the nucleotide (ADP) is moved outside the structure and written in the same color as the nucleotide is depicted in the model (similar as done in Fig. 4C).

Response: Following the reviewer’s suggestion, we moved the designation of ADP to outside the structure and used the same font color as the nucleotide model.

Fig. 1a: PD and TM need also aa residue numbers.

Response: Added. PD, periplasmic domain; TM, transmembrane.

Fig 3: Legend, line 6: (apo-sate) should read (apo-state)

Response: Fixed. Thanks.

Supplementary Fig. 4 legend: Here it is not clear which models were used for comparison. The pdb numbers and references of the used models have to be stated (also in other figures where relevant).

Response: The crystal structure used for comparison is the ADP-bound FtsH intracellular domains (PDB code 2CEA). We have added the pdb code in the figure legend. (p. 17, line 4)

Reviewer #2 (Remarks to the Author):

Summary:

The authors have solved the structure of *Thermotoga maritima* (Tm) FtsH in an ADP-bound state using cryo-EM. The cryo-EM imaging was conducted using full-length TmFtsH, and the authors solved the ADP-bound structure comprising hexameric ATPase and protease domains. Through a careful selection of 2D classes containing not only the ATPase and protease domains but also the transmembrane and periplasmic domains, the authors were also able to solve the structure of the ADP-bound full-length TmFtsH with low resolution. By comparing their ADP-bound structure with apo- or ATP-bound structures, conformational changes in the ATPase domains, rather than in the protease domains, were observed. When

considering the structure of the full-length TmFtsH, the authors found tilted topologies of transmembrane and periplasmic domains with extra densities in the large gap formed by the tilting, suggesting the mechanism of substrate recognition and initial loading. However, the biological relevance of this tilted structure has not been well supported, and the extra densities in the tilted region have not been fully described, even in the figures. Additionally, while the authors mentioned that the conformational changes in the ATPase and protease domains begin on the connecting loops, there is no detailed description provided. Although the manuscript is well written and easy to follow, the authors need to address the specific concerns listed below.

Response: We thank the reviewer for the insightful comments to help us improve the manuscript. In the revised manuscript, we have addressed these concerns point-by-point (please see responses below).

1. Throughout the manuscript, the description in the text and figures is not completely matched (e.g., on page 4, line 4, Fig 3c needs to be replaced with Fig 2c). In addition, there are several typos in the figure legends (e.g., in figure 2, even though there is no panel “g,” it is mentioned in the legend.).

Response: We have fixed these problems. Thanks!

2. Page 4, line 15: The authors have used a construct with H423Y mutation (a zinc-binding-deficient mutant) for cryo-EM analysis and have mentioned it as an active form in the discussion (page 7, line 5). Please show the additional data or provide pertinent explanation to support this.

Response: The H423Y mutant is protease deficient and could not cleave a protein substrate. However, its ATPase activity is reserved. This H423Y mutant has been reported to capture multiple FtsH substrates (Westphal et al. 2012, doi:10.1074/jbc.M112.388470). We attempted to add ATP-Mg²⁺ with the H42Y mutant to form an FtsH-ATP complex. However, in the reconstructed map, we observed only fully bound ADP molecules in the ATPase domains. We think that ATP was hydrolyzed during our sample preparation. We have revised the text and added the reference in the discussion to make it clear about the use of the protease deficient mutant. (p. 8, lines 9-10).

3. On page 5, line 1, the authors have mentioned that side chain densities were not observed for Arg318', however, the side chain of Arg318' has been marked in figure 3f. In addition, the authors have proposed a potential interaction between Arg318' and phosphate groups in ADP, but no pertinent evidence, such as a distribution of charges in the neighboring region or additional densities around the phosphate groups, has been provided.

Response: Due to the absence of side-chain density, the sidechain of R318' was modeled using its rotamer but only its C α atom position was used for distance measurement. Nevertheless, the densities for the R318' mainchain and ADP phosphate groups (Fig. 3f) can be observed. In the revised Fig. 3f, we have added mainchain densities (orange) for the R318' segment. (Fig. 3 and its caption)

4. Page 5, line 28-29: How does this structure provide the basis for substrate loading and translocation processes?

Response: Thanks for pointing out this overstated hypothesis. We have deleted this statement in the revised text.

5. In “Apo-ADP state transition” and “ADP-ATP state transition” sections, the authors have mentioned that the structural changes begin on the loops connecting the protease and ATPase domains. However, there is no description of how the structural changes in the loops are propagated to the ATPase domain.

Response: The connecting loops did not trigger the conformational changes in the ATPase domains. Instead, it's a consequence of conformational changes in the ATPase domains relative to protease hexamer. We have revised the text and added two supplementary figures (Fig. S4a, b) to show conformational changes in the ATPase domains. (p 5, lines 1-3; p 5, lines 26-28).

6. In "Full-length FtsH structure" section, the authors have introduced the full-length structure of TmFtsH whose ATPase and protease domains are tilted relative to the lipid bilayer. Is this tilted structure biologically relevant?

Response: We think this tilted structure is biologically relevant. Structurally, this tilted structure could be of relevance in allowing proximity of substrates to the ATPase pore loops, thus promoting substrate recognition. We have revised the text to include a discussion on the biological relevance of the titled structure. (p. 7, lines 24-25).

7. In the same section, the authors have suggested that the extra densities in the large gap formed by the tilting could originate due to the substrate or the disordered N-terminus of the protein itself. However, considering the low resolution of the full-length structure, especially in the transmembrane domain, it is not clear whether it is reasonable to assume extra densities. It is recommended to provide an additional figure with a description regarding the extra densities.

Response: Taking the reviewers' suggestion, we inspected the extra densities carefully. We agree with the reviewer that due to the disorder of the transmembrane TM domains and the connecting loops between the TM and the ATPase domains, it's premature to assume extra densities arising from a substrate. We have revised the text accordingly.

8. Page 7, line 8: What is the supporting data for this hypothesis?

Response: We deleted this hypothetical sentence in the revised Discussion section.

9. The authors have used the AlphaFold model structure to fit low-resolution densities in the full-length structure. However, it is unclear how the authors created a hexameric model structure using AlphaFold. Since the AlphaFold-Multimer was released recently, it would be helpful to mention in the method section, which version was used for the modeling and how it was carried out (AlphaFold, AlphaFold-Multimer, AlphaFold with modifications, or AlphaFold with Colab, etc.).

Response: The hexameric model was produced using ColabFold, a modified version of AlphaFold. We have added such information in Methods and cited the ColabFold bioRxiv paper (p. 22, lines 10-11).

Minor Concerns:

1. Please systematize the usage of the abbreviated letter code for amino acids.

Response: We have revised the manuscript throughout to use a single letter for amino acids.

2. Page 4, 2nd paragraph: It would be better to match the flow of the text and order of figures for easy understanding.

Response: We removed the out of order Fig 3f from the text.

3. Figure 1a: Please provide the explanation for PD and TM in the legend.

Response: Added. PD, periplasmic domain; TM, transmembrane. (p. 13, line 5)

4. Figure 4: The meaning of cyan color should be noted.

Response: Both cyan and green colors are for the ATP-state structure (Fig. 4a). We revised the figure and used a single color (i.e. green) for the ATP-state structure.

5. Figure 5e: Please add the explanation for Purple colored arrow in the legend.

Response: The purple-colored arrow was used to indicate the location of the extra densities. Taking this reviewer's suggestion, we dropped this premature claim of the extra densities and deleted the arrow in revised Fig. 6e (old Fig. 5e).

6. Supplementary Figure 1: Please add symmetry information for the refinement of the structure.

Response: Added.

Reviewer #3 (Remarks to the Author):

The authors provide a short report of the cryoEM structure of the ATP-dependent protease *Thermotoga maritima* FtsH to 3.15 angstrom resolution. FtsH is a membrane-anchored protease that can degrade both membrane and soluble proteins, and the authors use in vitro reconstitution into lipid nanodiscs to isolate and characterize the full-length protein in the fully-ADP bound state. This resolution is a significant improvement on a recently published study (Carvalho et al, 2021) visualizing detergent solubilized full-length FtsH. Previous cryoEM structures of mitochondrial homologues of FtsH reported highly asymmetric interactions with both nucleotide and trapped polypeptide substrates. Here, the authors reveal a largely symmetric structure with all six subunits bound to ADP.

The visualization of the fully-ADP state is of moderate interest to researchers in the AAA+ enzyme field and the high resolution of the reconstruction will add another group of states that are available to be occupied by these enzymes. However, it is unclear under what circumstances the enzyme will occupy this fully-ADP state. The authors suggest this may represent a 'resting state' of the enzyme when ATP is not available, but it would be helpful if they could provide some evidence from the literature regarding the relevance of this state under physiological conditions.

The previous cryoEM structures of mitochondrial FtsH-like enzymes lack the transmembrane domains and small domains that sit on the other side of the membrane. In that regard, information on the structure of these domains in the full-length enzyme is useful, albeit these domains are flexible and poorly visualized in this structure. A higher resolution reconstruction of these regions would have made this structure far more interesting.

The microscopy is well-performed, and the validation of the structure is sufficient.

Response: We appreciate the reviewer's overall comments and suggestions. We obtained the fully ADP-state structure with no special treatment during protein purification, reconstitution, and cryo-EM sample vitrification. That is, ADP was co-purified from the bacteria with the protein. The full-ADP bound structure has also been reported in the crystal structure of TmFtsH (PDB code 2CEA) although in that structure only the intracellular protease and ATPase domains were used for crystallization (Bieniossek, et al, 2006 doi:10.1073/pnas.0600031103). However, the crystallization of the intracellular domains resulted in a symmetry-break ADP-bound structure.

We agree with the reviewer that it will be far more interesting to have a higher resolution reconstruction for the transmembrane domains and small domains that sit on the other side of the membrane. However, to stabilize the transmembrane domains, one must screen for stabilizers including trying different membrane protein substrates. To assure proper substrate

engagement, proper amount of ATP and Mg²⁺ need to be screened. To slow down the ATP hydrolysis reaction to assure an engaged state, ATPase-attenuated mutants need to be produced and tested. The entire work will take a very long time to implement and thus will be a fascinating follow-up project to our work reported here.

Specific points

1. The authors state that they applied 6-fold symmetry to the reconstruction and that an alternative reconstruction lacking any applied symmetry had slightly lower resolution. How do the positions of the ATPase domains in these two reconstructions compare? Is there evidence of asymmetry in the unconstrained map that is being lost when 6-fold symmetry is applied?

Response: We compared the two maps reconstructed with C6 and C1 symmetry. However, we did not observe any evidence for asymmetry in the C1 map.

2. The manuscript contains a number of spelling and grammatical errors. It would be improved by a careful revision to remove these errors.

e.g. --Figure 1e is mislabeled in the legend.

--Page 3 Line 31 "followed" should be "following"

--Page 4 line 7 "much few" should be "many fewer"

Response: We have gone through the text carefully and fixed these problems. Thank you.

Reviewer #4 (Remarks to the Author):

This paper reports a cryo-EM study of AAA+ protease TmFtsH in ADP-bound states, including a low-resolution reconstruction of the full-length complex. While the work appears to be technically sound, the biological insights obtained are limited and incremental. Particularly, there is lack of biochemical validation of the purified and imaged complexes, thus leaving it to be less well-defined in terms of the functional state of the solved structure. With additional verification and control, the authors indicated that these structures represent the resting state, which is likely but still hypothetical, and weakly discussed the possible meaning of the structural model and how it adds to our knowledge to the system. In this reviewer's opinion, the paper may be enhanced at least by running a few basic assays to test the protease function. If possible, it will greatly help with a few control cryo-EM reconstructions with ATP or substrate added, which would make the structural interpretation and insights more reliable. At the current situation, one can only make bold assumption and vague discussion as to what has happened to the complex and what we can learn from the symmetric structure, when nearly all substrate-bound AAA+ ATPase complex showed asymmetric conformations. Additionally, the authors might want to consider the following issues when revising the paper.

Response: We thank the reviewer for their insightful comments on our work. The protein that we used for cryo-EM work is a H423Y protease deficient mutant. The mutant has been used by others to trap multiple FtsH substrates from bacteria (Westphal et al. 2012, doi:10.1074/jbc.M112.388470). Due to inactivity on protein cleavage, we did not use the H423Y for protease assay. We did attempt to incubate 5-10 mM ATP and 5 mg Mg²⁺ with the FtsH-MSP nanodiscs for obtaining an ATP-state structure. However, in the reconstructed map we observed only the full ADP-bound structure. It's possible that the ATPase activity of the H423Y mutant converted ATP to ADP during our cryo-EM sample preparation. Please see our revised text in Discussion section. (p. 8, lines 5-10).

We agree with the reviewer that the current work will be enhanced by forming a complex with a substrate. However, the conditions required to form a suitable complex for cryo-EM work is unknown. It may take a lot of time in search of substrates and lipid compositions. As pointed

out by the reviewer, nearly all substrate-bound AAA+ ATPase complexes have ATP bound. So, it's necessary to find and produce an ATPase-attenuated *TmFtsH* mutant in order to capture its substrate by the ATPase pore loops. The transmembrane domains in *TmFtsH*, which are not present in all other AAA+ ATPase complexes, may further increase the complexity in both complex formation and cryo-EM sample preparation. We conclude that the reviewer's suggestion may be better addressed by a follow-up project to determine a *TmFtsH*/substrate complex.

(1) The authors need to clarify the biochemical condition with respect to the nucleotide type and concentration used in the structural study both in the first section of results and in the Methods. Were the full-length *TmFtsH* purified with the presence or the absence of ADP or ATP in the buffer? Any magnesium ions provided in the buffer? If no, explain the rationale for why no nucleotides were supplied for the complex?

Response: To obtain an ADP-bound structure, adding nucleotides is not necessary as demonstrated in the ADP-bound crystal structure of the *TmFtsH* intracellular domains (PDB code 2CEA). In both crystal (PDB code 2CEA) and cryo-EM (this work) structures, ADP was co-purified with the protein from the bacteria. So, we did not add ADP, ATP, or magnesium ions in the buffer for purification and MSP nanodiscs formation. We have added detailed biochemical conditions to the first section of results (p. 4, lines 20-23) and the Methods (p 19, lines 29-30; p 20, line 23).

(2) Where do the ADP ligands bound come from? Were they bound endogenously and copurified or added in the late step of sample preparation?

Response: ADP ligands bound endogenously to *TmFtsH* and survived the co-purification MSP nanodiscs reconstitution processes. (p. 4, lines 22-23)

(3) What were the nucleotide states of the full-length *TmFtsH* reconstruction shown in Fig. 5?

Response: The low-resolution reconstruction at 7.9 Å does not support the identification of nucleotides in the reconstructed map.

(4) Have the authors conducted degradation assay and/or ATPase activity assay to verify the biochemical activity and function of FtsH or FtsH-MSP nanodisc?

Response: We obtained only a minute amount of FtsH-MSP nanodiscs for cryo-EM work. We haven't done ATPase assays with the FtsH-MSP sample. Nevertheless, we did attempt to prepare an FtsH/ATP-Mg²⁺ complex. However, in the reconstructed map, we observed only fully bound ADP in the ATPase domains in the same 6-fold symmetry as the ADP-state structure. (p. 8, lines 5-12)

(5) The discussion of comparison of the ADP-bound cryo-EM structure with those of ADP-bound crystal structures is not clear. There appears to be notable differences in Supplementary Fig. S4. However, no discussion on such structural differences is provided and any explanation on the compatibility of cryo-EM and crystal structures, as well as why the two methods yield different conformations of the same nucleotide states. This concerns whether the current study rectifies the previous crystal structure study and whether the presented structure implicates a different mechanism.

Response: We thank the reviewer for pointing out this discrepancy. Indeed, our cryo-EM structure rectifies the crystal structure which used only the soluble intracellular domains (ATPase and protease) for crystallization and structure determination. To give a more detailed comparison of the cryo-EM and crystal structures, we renamed the old Fig. S4 to be a new Fig. 5 and added a section to compare the two structures. (p. 6, lines 10 - p.7, line 3)

(6) Can the authors comment on the possible reason why all six subunits are in symmetric ADP-bound state? It also seems to be a simple test if the authors can add ATP to the purified complex to see if their assertion that ATP and substrate binding are needed to break the symmetric is correct.

Response: We could hypothesize that the binding of one ADP molecule to a symmetric FtsH hexamer would bring the next clockwise subunit closer, induce conformational changes, and break symmetry. In presence of sufficient amount of ADP, additional ADP binding will trigger conformational changes in the next clockwise subunit until all six subunits reach a new symmetric ADP-bound state.

We did try to include ATP to form a symmetry-break complex. However, we ended up with only ADP in the ATPase domains. Obtaining a substrate-bound cryo-EM structure is not trivial for FtsH (see our response to the overall comments) and warrants a follow-up project. (p 8, lines 5-12)

(7) While the authors claim in the abstract that a possible mechanism of substrate recognition and loading, there are no well-phrased passage describing such a mechanism, except for a few sentences speculating highly hypothetical, vaguely conveyed possibilities. These may not be sufficient to support their claim in the abstract.

Response: We have revised the abstract and deleted this hypothetical statement.

REVIEWERS' COMMENTS:

Reviewer #2 (Remarks to the Author):

The authors have responded to all my questions and made the necessary changes to the manuscript.

Reviewer #4 (Remarks to the Author):

The authors have addressed all my questions. I recommend its publication in the present form.

REVIEWERS' COMMENTS:

Reviewer #2 (Remarks to the Author):

The authors have responded to all my questions and made the necessary changes to the manuscript.

Response: We thank the reviewer's effort and time to help us improve the manuscript.

Reviewer #4 (Remarks to the Author):

The authors have addressed all my questions. I recommend its publication in the present form.

Response: We thank the reviewer's effort and time to help us improve the manuscript.